# Novel Treatment Approach for Aspergilloses by Targeting Germination

**DOI:** 10.3390/jof8080758

**Published:** 2022-07-22

**Authors:** Kim Verburg, Jacq van Neer, Margherita Duca, Hans de Cock

**Affiliations:** 1Molecular Microbiology, Department of Biology, Faculty of Science, Utrecht University, Padualaan 8, 3584 CH Utrecht, The Netherlands; verburgkim@gmail.com (K.V.); j.f.vanneer@uu.nl (J.v.N.); 2Department of Chemical Biology & Drug Discovery, Utrecht Institute for Pharmaceutical Sciences, Utrecht University, 3584 CG Utrecht, The Netherlands; m.duca@uu.nl

**Keywords:** *Aspergillus*, aspergillosis, *Aspergillus fumigatus*, azole resistance, conidia, cystic fibrosis, germination, lung infection

## Abstract

Germination of conidia is an essential process within the *Aspergillus* life cycle and plays a major role during the infection of hosts. Conidia are able to avoid detection by the majority of leukocytes when dormant. Germination can cause severe health problems, specifically in immunocompromised people. Aspergillosis is most often caused by *Aspergillus fumigatus* (*A. fumigatus*) and affects neutropenic patients, as well as people with cystic fibrosis (CF). These patients are often unable to effectively detect and clear the conidia or hyphae and can develop chronic non-invasive and/or invasive infections or allergic inflammatory responses. Current treatments with (tri)azoles can be very effective to combat a variety of fungal infections. However, resistance against current azoles has emerged and has been increasing since 1998. As a consequence, patients infected with resistant *A. fumigatus* have a reported mortality rate of 88% to 100%. Especially with the growing number of patients that harbor azole-resistant Aspergilli, novel antifungals could provide an alternative. Aspergilloses differ in defining characteristics, but germination of conidia is one of the few common denominators. By specifically targeting conidial germination with novel antifungals, early intervention might be possible. In this review, we propose several morphotypes to disrupt conidial germination, as well as potential targets. Hopefully, new antifungals against such targets could contribute to disturbing the ability of Aspergilli to germinate and grow, resulting in a decreased fungal burden on patients.

## 1. Introduction

The *Aspergillus* species is one of the most common invasive fungal pathogens, as they make up about 20% of the invasive fungal infections in the US [1]. According to current estimates, more than 300,000 life-threatening cases of invasive pulmonary aspergillosis (IPA) occur annually worldwide, with a reported mortality rate of 30 to 95% [2,3]. As displayed in Figure 1, aspergillosis comprises numerous diseases caused by *Aspergillus* species, such as non-invasive aspergillomas, allergic bronchopulmonary aspergillosis (ABPA), chronic pulmonary aspergillosis (CPA), chronic necrotizing pulmonary aspergillosis (CNPA), IPA, severe asthma with fungal sensitization (SAFS), and extrapulmonary aspergillosis [4,5,6]. *Aspergillus fumigatus* (*A. fumigatus*) causes the majority of aspergillosis cases [7,8]. The main cause of IPA is *A. fumigatus* (with 57%), while other species, such as *A. flavus*, *A. niger* and *A. terreus* were found in about 12%, 10% and 12% of the cases, respectively, although these numbers can vary between countries [4]. In addition, infections with cryptic species of *Aspergillus* have been increasingly identified in clinical settings in the last few decades [9,10,11,12,13]. What is more, Aspergilli reside on crops such as nuts, grains, beans, and dried fruits, which leads to spoilage [14]. However, the presence of Aspergilli on crops can be harmful to humans because they exudate mycotoxins, including aflatoxins [15]. Aflatoxin-contaminated foods were found to impact health negatively, considering that aflatoxin was found to play a major role in 4.6 to 28.2% of global liver cancer cases [16,17]. Clearly, the impact Aspergilli has on human health is extensive, but in consequence of the major mortality rates and the shortcomings of current azole-based therapies this review will focus on the pathogenicity.

Daily, humans inhale a relatively low amount of *Aspergillus* spores estimated at around 100–1000 spores, which they can eliminate from their lungs without the occurrence of inflammation [18,19]. Aspergilli have conidiophores that produce and release conidia, which are asexual spores [20,21]. About a third of Aspergilli, including *A. fumigatus*, have a sexual life cycle, during which cleistothecia develop and produce asci, which produce sexual spores called ascospores [21,22,23,24]. *Aspergillus* conidia can be found in the air as bioaerosols, both indoors and outdoors. As such, they pose an eminent risk for immunocompromised and/or immunodeficient people, and also when in high concentrations for the healthy population as well [1,25]. Conidia are small and hydrophobic, characteristics predisposed to facilitate dispersion into the air [26,27]. The increased efficiency of dispersal of conidia in the air by *A. fumigatus* compared to other Aspergilli contributes to their higher prevalence and infection rates [8]. What is more, the conidia have a high stress resistance and the ability to germinate at 37–39 °C [26,28,29]. This makes Aspergilli resilient and highly suited to the conditions of human hosts.

Specific groups of people with underlying (immune) diseases experience specific aspergilloses more often. It was found that chronic pulmonary aspergillosis (CPA), a chronic *Aspergillus* infection, is more common in patients with damaged lung tissue and/or pulmonary diseases [30,31]. Pulmonary tuberculosis (PTB) is one of the most common diseases worldwide, with about 9 million cases each year. Especially when left untreated, patients often develop lung damage and, as a consequence, have a higher risk of developing CPA [32]. Similarly, ABPA was thought to impact only 17.7% of patients with cystic fibrosis (CF) but that may be a gross underestimation, as it is now estimated that almost 50% of adult CF patients have/have had ABPA or *Aspergillus* bronchitis (AB) [33]. Infections caused by *Aspergillus* species were most commonly found in patients with CF [34,35,36,37,38], chronic obstructive pulmonary disease (COPD) [2,30,39,40], transplant recipients [41], hematopoietic stem cell transplant (HPSC) recipients [4,42,43,44,45], and especially neutropenic patients [46,47]. The common denominator is that the immune system is compromised and thus unable to recognize and remove the (dormant) conidia and developing structures [3,48,49]. The presence of cell wall debris and/or hyphal fragments, either dead or alive, is sufficient to trigger immune activation in such a way that it can cause asthma and hypersensitivity pneumonitis [50]. It should be mentioned, however, that bronchial epithelial cells are part of the defense and were shown to process dormant conidia by clearance from the epithelium, thereby preventing germination [51]. In addition, bronchial epithelial cells can process dormant conidia via uptake and killing [52], although various studies showed that these DHN-melanin-containing conidia can escape being killed by suppressing phagolysosomal acidification [53,54]. Excess mucus formation in CF patients affects, among other things, mucociliary clearance and, in addition, prevents interactions with bronchiolar epithelial cells, thereby compromising conidial clearance [55,56]. It was also found that up to 60% of germlings internalized by epithelial lung cells are able to escape phagosomes through dysregulation of multiple processes because the phagosomes do not acidify, and hyphal structures escape the phagosomes without undergoing lysis [40]. Therefore, Aspergilli evade the Th1 response that effectively clears the conidia in healthy patients [57]. Consequently, the fungus can colonize the host tissue [47].

Since the current COVID-19 pandemic started in 2019, co-infection of SARS-CoV-2 and *Aspergillus* is becoming prevalent [58,59,60]. What is more, the frequency of co-infection with IPA ranged from 19.6 up to 33% of patients but occurred mainly in severe COVID-19 cases. Subsequently, several hospitals have decided to use antifungal medications such as amphotericin B (AmB) and azoles as a prophylactic treatment for every intubated and mechanically ventilated patient in the ICU [61]. This poses a quandary, as drug-resistant Aspergilli are becoming more prevalent, and they cause a higher mortality rate than their non-resistant counterparts [62]. A recent study in the Netherlands has found that of the 31.7% of CF patients in CF centers that were infected with *A. fumigatus*, 7.1% had an azole-resistant strain [63]. Especially as current first-line treatment includes mostly (tri)azoles. Subsequently, the emergence of (tri)azole-resistant species, markedly *A. fumigatus*, is having a highly negative impact on patient survival and health [64,65].

With the emergence of more resistant *A. fumigatus* in patients, the development of new antifungal drugs is expedient for the treatment of aspergillosis. Further alternatives include cathelicidins, which are antimicrobial peptides such as LL-37, which is also used by mammalian cells [29]. New targets must be identified, and recent transcriptomic and proteomic studies could be of use to uncover potential targets [39,48]. Some potential targets for novel antifungal drugs are transcription factors such as Crz1, RlmA, SltA, AtfA, and BrlA [66,67,68,69]. Other targets include stress–response components of *Aspergillus*, such as trehalose [69]. Considering the role of germination regarding the pathogenicity of *A. fumigatus*, antifungals with the aim of precluding germination would be promising [3,40,70,71]. More advances can be made with an expanded knowledge of germination, azole resistance, and genomic and transcriptomic studies that focus specifically on germinating Aspergilli. As of now, there is a variety of proposed targets for new antifungals. The aim of this review is to sort out the currently proposed targets and uncover potential new anti-germination targets in *A. fumigatus*, considering the properties of antifungal drugs and the arising resistance.

## 2. Host Susceptibility—Immune Clearance and Evasion of the Immune System

Predisposition to aspergillosis is linked to diseases such as AIDS, chronic granulomatous disease (CGD), CF, asthma, influenza [72], COVID-19, immune suppression due to the use of immunosuppressive drugs such as corticosteroids [73], and lung damage as a result of COPD and PTB [5,45,74,75,76,77]. Additionally, host genetics play a role in susceptibility, such as mutations in Dectin-1 and -2 [78] and STAT3 [79]. Disparate aspergilloses, such as IPA, CPA, SAFS, ABPA, non-invasive, extrapulmonary and acute/community-acquired, occur more frequently within specific patient groups, as can be seen in Figure 1 [4,75,80]. This diverse range of illnesses indicates a disparity in either the specific isolates that cause distinct aspergilloses or between the patients. Alternatively, both could play a role. Unlike other *Aspergillus*-caused diseases, CPA has an estimated burden of up to 3 million worldwide and impacts mostly immunocompetent people after pulmonary tuberculosis [6,32]. Nevertheless, generally, people with a compromised immune system suffer the most from aspergilloses [43]; What makes these immunocompromised patients more susceptible to different types of aspergillosis? An important aspect to consider is whether the pathogenicity varies between various isolates and how Aspergilli have evolved as both plant and animal pathogens. With more insight into what makes Aspergilli pathogenic and which conditions are most suitable for germination, this could be applied to develop new antifungal or anti-Aspergillus drugs. Especially with the growing burden of tuberculosis, COVID-19, and other lung- and immune-affecting diseases worldwide, the encumbrance of aspergillosis on the worldwide population will grow.

### 2.1. Pathogenicity of Aspergilli through Resilience and Dispersal

Several Aspergilli are highly virulent in humans, although some, such as *A. flavus*, were reported as plant pathogens as well [21]. Moreover, their resilience is displayed through their ability to grow under relatively extreme circumstances such as temperatures ranging from 6 to 55 °C, as well as with very little organic resources, and in low humidity [21,28,81]. In addition, *A. fumigatus* is versatile in its ability to use carbon and nitrogen sources, which enables the fungi to grow in a variety of environments due to its nutritional versatility [82]. Infections by Aspergilli, but especially *A. fumigatus*, are also associated with the ability to disperse conidia efficiently [8,19]. The combining factors of conidial size (2–3 µm), hydrophobins, and shielding cell wall structures all conduce deep penetration of the lungs as well as evasion of the host’s immune responses [18,83,84,85,86,87,88]. 

The conidia of Aspergilli have different cell wall layers, and the one that forms the outer layer is mostly comprised of melanin [89] and hydrophobins, especially RodA [88]. Under this cell wall layer, polysaccharides such as β-1,3-glucan, α-1,3-glucan, α-mannan, galactomannan (GM), and galactosaminogalactan (GAG; not present in dormant conidia) form an interconnected network with chitin up until close to the plasma membrane [90,91]. Conidial melanin is a pigment present in conidia that has a role in platelet activation [92]. Different melanin pigments were found in *A. fumigatus*, among which are DHN-melanin and pyomelanin [93]. It was found that *A. fumigatus* conidia have adapted to evade the host’s immune system by expressing hydrophobics, of which RodA appeared to be the most important because RodA is required for rodlet formation on the surface of the conidia [86,88,94]. When RodA is mutated, evasion of the immune system is disrupted, as a consequence of the disturbance in the rodlet layer [87]. The function of RodA in conidia is to prevent exposure of pathogen-associated molecular patterns (PAMPs), such as β1,3-glucan and α-mannan to the host’s immune system [85,87]. More recently it was found that CcpA is another conidial surface protein of dormant conidia that prevents recognition by the host’s immune system, similar to RodA [83]. After cell wall remodeling takes place during germination, CcpA maintains the functionality of contributing to immune evasion of conidia. This is not the case with RodA, as conidia relinquish RodA during germination. Due to the ability of dormant conidia to evade the host’s immune system with proteins such as CcpA and RodA, clearance of these (dormant) conidia can be hindered in healthy people as well in immunocompromised individuals. 

### 2.2. Normal Host Clearance of Aspergillus

Antigen-presenting cells (APCs), including most types of macrophages, can efficiently recognize PAMPs and present antigens to other immune cells. Consequently, either the Th1 or Th2 immune response is activated, depending on the production of signaling molecules such as cytokines and interleukins (IL), for example, tumor necrosis factor α (TNF-α) and interferon-gamma (IFN-γ) [80]. When healthy individuals come into contact with *Aspergillus* conidia, PAMPs are recognized by lymphocytes. As a result, TNF-α, IL-2, IFN-γ, and granulocyte–macrophage colony-stimulating factors are produced, while mononuclear cells (MNCs) present antigens that eventually lead to induction of the Th1 immune response [57]. Macrophages are monocyte-derived cells that express lectins, among others, which are imperative for various cellular recognition events. Specific C-lectins, Dectin-1 and Dectin-2 are essential in the recognition of *Aspergillus* conidia by macrophages, as well as the successive production of different ILs and TNF-α [85]. Conidial cell wall component DHN-melanin can be recognized by C-type lectin MelLec, which is mostly involved in recognition by myeloid and likely epithelial lung cells [95]. Internalization of dormant conidia by epithelial lung cells is mediated by EphA2 and Dectin-1 as a response to DHN-melanin and glucans exposed on the surface [96]. However, Aspergilli have partially adapted to evade recognition by the immune system with hydrophobic rodlets and proteins RodA and CcpA on the surface that conceal PAMPs when conidia are dormant [86]. What is more, the immune system of individuals with compromised immune systems is often unable to detect and act upon contact with *Aspergillus* conidia.

### 2.3. Interactions between Hosts and Aspergillus Conidia 

As the main cause of aspergillosis, *A. fumigatus* was found to respond to its host’s environment by expressing different genes at different time points [42,68,83,97,98]. Compounds that Aspergilli secrete, such as gliotoxins, have a considerable impact on mammalian cells through reactive oxygen species that mediate oxidative damage and apoptosis [99]. Intriguingly, *A. fumigatus* conidia have shown differential expression of nine genes encoding for proposed secreted proteins during host infection in two different murine models [42]. They found that there was a significantly different expression of *A. fumigatus* genes between days two and three of inoculation of steroid model and chemotherapeutic model mice when studying IPA. In steroid-treated model mice, 35 genes were upregulated on day three, while only 24 genes were upregulated in the chemotherapeutic model. Most of the transcriptomic differences were found to lie within the gliotoxin biosynthesis gene cluster, which was found to be downregulated in the chemotherapeutic model. Moreover, *A. fumigatus* can recognize and subsequently modulate gene expression upon encountering neutrophils from either immunocompetent people or neutropenic CGD patients [98]. Taken together, this indicates that Aspergilli, or at least *A. fumigatus*, behaves differently depending on the immune status of individual hosts. This specific adaptation is possibly why patients with underlying conditions experience disparate aspergilloses, especially if this means that Aspergilli infect immune deficient patients more frequently.

### 2.4. Abnormal Host Immune Responses to Aspergilli

Immunocompromised individuals have other responses to Aspergilli compared to healthy individuals, which is apparent in the case of ABPA and SAFS. In patients that experience ABPA, *Aspergillus* conidia induce relatively high levels of IL-4, IL-5 and IL-13, and low levels of IFNγ [100,101,102,103]. These interleukins are cytokines associated with a Th2 immune response, which includes stimulation of B-cells that produce a host of cytokines that lead to IgE antibody formation and inflammation [104,105]. Inflammation is a consequence of activation of mast cells and recruitment of eosinophils, mainly by cytokines such as IL-5 combined with high levels of IgE [101,106,107]. Patients that exhibit this type of immune response, have an impaired ability to eradicate *Aspergillus* fungal growth. Contrastingly, recognition of *A. fumigatus* by monocytes from healthy hosts led to higher expression of interleukins (IL-10, IL-8, IL-1β, CXCL2, CCL3, CCL4, and CCL20), CD14, matrix metalloproteinase 1 (mmp1), ficolin-1, and opsonin long pentraxin 3 [108]. With regards to the eosinophil stimulation and maturation that takes place upon the response of *Aspergillus*-antigens, targeting the IL-5 pathway might have some potential [109,110]. Considering the role of the individual cytokines and the balance of Th1 and Th2 immune responses, modulating this to the advantage of patients would likely need customization for patients or patient groups. 

Because of the high prevalence of allergic aspergilloses such as ABPA (10.5%) and SAFS in the specific risk group of CF patients, it might be promising to look into specific therapies [111]. The inflammatory response, which is a key symptom of ABPA, which primarily CF patients exhibit, could possibly be targeted through modulating TLR-4/TLR5 [112]. It was found that the cystic fibrosis transmembrane conductance regulator (CFTR), which causes CF when defective due to mutations, is important in the regulation of CD4+ T-cells. Consequently, mutations in CFTR lead to a bias towards a Th2 response to *A. fumigatus* antigen exposure, as IL-4, IgE and IgG are elevated compared to wild-type CFTR [49,112,113]. Co-infection of Aspergilli and *Pseudomonas aeruginosa* (*P. aeruginosa*) might contribute to the prevalence of ABPA in CF patients, as *P. aeruginosa* mediates a shift towards the Th2 response [114] and is found in CF patients with a prevalence of about 26.9% [115]. The presence of *P. aeruginosa* likely contributes to higher host susceptibility to aspergillosis, specifically through a TLR5-mediated shift to a Th2 inflammatory response concordant with ABPA symptoms [100,116]. In addition, a new line of therapy to treat CF was introduced, which corrects effects caused by mutations in CFTR by modulating CFTR [37,117,118,119]. Considering this, forestalling a TLR5-mediated shift, combined with prophylactic inhibitory drugs that target Aspergilli, could possibly prevent the colonization of *A. fumigatus* before the development of ABPA. 

## 3. Current and Past Treatments for Aspergillosis and the Rising Azole Resistance

The first antifungal drugs that were used for aspergillosis are amphotericin B (AmB) and itraconazole, as mentioned in Table 1. AmB binds to sterols, leading to increased permeability of membranes and inhibition of ATPase proton pumps [19]. However, AmB has been used for more than 50 years and different formulations such as L-AMB have been developed, but there are also some drawbacks, even though it is effective for treating invasive aspergillosis [120]. For one, it is hydrophobic and thus difficult to administer to patients affected by aspergillosis. Most detrimental is the toxicity of AmB on patients, even though fungal cells are affected more than host cells. Itraconazole is a triazole that binds competitively to a catalytically active iron atom in the enzyme14α lanosterol demethylase, which is involved in cytochrome P450 (CYP450) [19,121]. Since 1998/1999, resistance against azole medications emerged among patients with aspergillosis, first reported in the Netherlands [122]. The isolates were specifically resistant to itraconazole, but not the newer voriconazole medication, which had become the new first-line treatment since 2002 [123]. Following that, other azole-based antifungals such as posaconazole, as well as echinocandins caspofungin, micafungin and anidulafungin were used as (invasive) aspergillosis treatments [62,124,125,126]. Most of these azoles have a similar mechanism of action; affecting sterols in the plasma membrane by targeting the 14α lanosterol demethylase enzyme, thereby also targeting the CYP450 family. It should be noted that new antifungals are under development and target, for example, GPI-anchor biosynthesis or sphingolipid synthesis [127], as well as cell wall synthesis and remodeling [128]. As of 2009, azole resistance became more common as from 6 to 12.8% of patients in the Netherlands were found to have resistant isolates, yet these numbers may vary between countries [129,130]. The development of isavuconazole, a newer azole-based antifungal was to no avail [131,132,133]. This was because the isolates resistant to itraconazole showed a reduced susceptibility for voriconazole and posaconazole [129,130]. Currently, azole resistance is still a problem for patients with aspergillosis, especially immunocompromised patients. Among patients, aspergillosis leads to a further decline in respiratory health, and the azole resistance of *A. fumigatus* was found in 7.1% of the culture-positive CF patients [63]. 

### 3.1. Mechanisms of Azole Resistance

Most of the azole-resistant strains have mutations in the *cyp51A* gene, which is part of the CYP450 family, with hotspots at codons 54, 98, and 220 [134,135,136]. However, not only mutations in the *cyp51A* or *cyp51B* genes [137] can lead to resistance, but mutations in other genes such as *afcox10*, *hapB*, *hapE*, *hmg1*, *mfsC*, *nctA*, *nctB*, *srbA*, *and svf1* were identified to contribute to azole resistance as well [138,139,140]. The most dominant mutations, especially for multi-azole resistance, were tandem repeat (TR) mutations of 34 bp (also called TR34) and an L98H substitution in *cyp51A* [141]. Tandem repeat mutations in the promotor region such as TR34 or TR46, lead to increased expression of *cyp51A* and combined with the L98H mutation, contribute to a multi-azole resistant phenotype [129,142]. *Cyp51A* encodes for the enzyme 14α-lanosterol demethylase which is essential for CYP450 functioning. Prevalent mutations that were found in azole-resistant isolates include L98H, TR34, relocation of tyrosine 107 and 121, and G54W, which all led to a smaller binding site, resulting in impaired binding of (tri)azoles [141]. What is more, no unfavorable fitness costs were found with these azole-resistant mutations. This is most likely because the mutated Cyp51A protein functions normally, even though the mutated regions are normally highly conserved [143]. It was proposed that specific mutations lead to resistance to either multiple azole resistance or to resistance for specific azoles. For example, a G448S mutation in *A. fumigatus* in *cyp51A* was associated with resistance against voriconazole, but not posaconazole [144]. It seems to be the case that azole resistance is a broad term that describes several mutations that can be obtained, but mostly (up to 80% of azole-resistant strains) in the *cyp51A* gene, with the most common mutations being TR34 and L98H in A. fumigatus [63,145].

**Table 1 jof-08-00758-t001:** Common antifungal drugs used as aspergillosis treatment. Itraconazole and fluconazole were part of the first generation of azole drugs. Second generation of azoles include voriconazole, posaconazole, and isavuconazole. CYP450 as target means that the ergosterol synthesis is affected, which is essential for the cell membrane. This is performed by inhibiting 14α lanosterol demethylase, on which CYP450 depends.

Antifungal Drug	Drug Target	Year Dispensed	Resistance Status	Source(s)
**Amphotericin B (AmB)**	Sterols in the membrane of Aspergilli. Increased permeability and inhibition of ATPase proton pumps	1958 (re-introduction in 1990s with lipid-based AmB)	Yes, but uncommon	[19,62]
**Itraconazole**	CYP450	1992	Yes	[121]
**Caspofungin**	Synthesis of cell wall component 1,3-β-d-glucan	2001		[121,125]
**Voriconazole**	CYP450	2002	Yes	[121,123]
**Micafungin**	Synthesis of fungal cell wall component β-1,3-glucan	2005	Yes	[126]
**Posaconazole**	CYP450	2006	Yes	[124]
** *Anidulafungin* **	Synthesis of fungal cell wall component β-1,3-glucan	2006	Yes	[146]
**Isavuconazole**	CYP450	2015	Yes	[131,132,133]

The antifungal drug names are in bold; the fields on the right are colored to indicated the correct parts belonging to the drug described.

Other prevalent *cyp51A* mutations were found in patients with aspergillosis, including a less prevalent 53bp TR mutation (also called TR53) [147] and a 46 bp TR mutation (also called TR46) with Y121F and T289A mutations [148]. Moreover, in this study, most of the patients that harbored the TR46/Y121F/T289A mutations were diagnosed with (probable) IPA and had not undergone azole treatments prior to the infection [148]. About half of the patients with probable or proven IPA in the study received voriconazole as first-line treatment and all died within 12 weeks of culturing of the isolate. The other half received liposomal AmB, and those patients all survived until at least 12 weeks, although two of the patients maintained a persistent infection. These findings highlight that the mutation was already present in the fungi and that, at least in these cases, the first-line treatment solely affected patient survival but not resistance. 

### 3.2. The Rise and Challenge of Azole Resistance

It is highly sought after how azole resistance has originated, as azoles have now been used for over 20 years in the medical field as well as in agriculture [122,129]. Azole-resistant Aspergilli were found in environmental samples/isolates in the Netherlands, Belgium, India, Italy, and the UK [145,148,149,150,151]. Furthermore, high levels of azole-resistant *A. fumigatus* were found in a study on azole resistance (mutations) in organic waste from landscaping, flower bulb waste, wood chippings, and household waste [152]. These findings are in line with earlier findings that related azole-resistant strains to the isolates that caused IPA could be found in air samples in the surrounding environment [148] and which was confirmed by population genomics analysis [145]. More comparable azole resistance genetic markers were found in patients with aspergillosis or *Aspergillus* colonization and environmental samples of Aspergilli [145]. Additionally, it was found that the agricultural use of azoles puts enough selective pressure on *A. fumigatus* for it to acquire the TR34/L98H mutation to induce azole resistance [153,154]. Similar results for azole-based fungicides were found for G54 mutations, among others [155,156]. This is despite the regulation of azoles and other fungicides in agriculture and the different azoles (DMIs) that are used in agriculture compared to first-line treatment for patients. The urgency of the effects that the usage of azoles has in agriculture was highlighted by findings of a novel pan-triazole-resistant mutation with a triple TR46-bp promoter repeat in the promoter region of *cyp51A*, possibly due to sexual reproduction [157]. In-patient acquired resistance was found as well, especially in patients with CPA and ABPA, or those with reoccurring or persistent IPA infections [158,159]. The (multi-)azole resistance in *A. fumigatus* has an exceptionally detrimental effect on patient survival, as the mortality of patients infected with resistant strains was found to be between 50 and 100%, although some have reported that it would be between 88 and 100% since the transition from the “azole era” to the “azole-resistant” era [65,148,160].

## 4. Conidial Germination and Its Different Morphotypes

Aspergilli are a diverse group of fungi, currently comprising 446 species, and all of them go through an asexual life cycle where conidia are produced [23,161]. Conidia are resistant to many stressors such as temperature shocks, drought, low nutrient availability as well as osmotic and oxidative stress [21,162]. This results from the physical barrier consisting of layers of polymers that encompass conidia, molecules such as trehalose and mannitol, as well as the ability of conidia to stay dormant [162,163,164]. Germination of *A. fumigatus* was shown to be heterogeneous and conidia program themselves by the formation of transcripts during sporulation, which is affected by culture conditions [165,166,167]. Furthermore, heterogeneity between conidia in cell wall composition affects fungal sensitivity and phagocytosis [168]. During germination, conidia have distinct morphotypes, which can be seen in Figure 2. Resting or dormant conidia of *A. fumigatus* make up the first of the germination morphotypes, where they are the smallest with a diameter of about 2–3 µm [19]. In this dormant morphotype, conidia can disperse, evade the immune system of their host, and withstand stressors until the environment is favorable for germination [28,169]. The second morphotype is the breakage of dormancy, which is not visible but characterized by transcriptional changes in the conidia [170]. Early conidial germination and the timing of the transitions are dependent on the availability of resources, such as inorganic salts, sugars, and amino acids [27,28]. In the third morphotype, isotropic growth takes place, where conidia swell to about double their original size [170]. The fourth morphotype is when a switch from isotropic to polarized growth takes place. A germ tube, otherwise known as a polar tube, is formed during polarized growth where the outer spore wall is breached [171]. When these germ tubes continue to grow, vegetative growth takes place, where the formed hyphae grow. 

### Transitions and Differences between Morphotypes; How Are They Facilitated?

Transitions between each of the morphotypes of germination are highly regulated and were studied at the RNA level [170]. The functions associated with the transcripts were distinct for the conversion between the morphotypes. In dormant conidia, there is mostly RNA encoding for polymers and other shielding- and preservation-associated (glyco) proteins important for the synthesis of the cell wall, as well as enzymes and other (secondary) metabolic pathway components [173].

In a study, *A. niger* was found to express different genes only 1 h after inoculation of newly harvested dormant conidia in a medium [174]. They found that upregulated genes were associated with fermentation/glycolysis, nitrogen metabolism, TCA cycle, mitochondria, and respiration. Whereas genes that were downregulated were associated with the metabolism of internal and alternative carbon sources, the GABA metabolic pathway, and anaerobic respiration mediated by glyoxylate. This differs compared to a similar study of *A. niger* conidia where transcripts upregulated after breaking dormancy (after 2 h) were associated mostly with the cell cycle, DNA processing, respiration, and protein synthesis [170].

Some of the genes associated with internal carbon source metabolism are mannitol dehydrogenases such as *mpdA*, glycerol dehydrogenase, glyceraldehyde dehydrogenase *gpdA*, and trehalase (*treB*). An abundance of transcripts was found in dormant conidia, as well as during isotropic and polarized growth, but not shortly before isotropic growth takes place [174]. One of the most important differences found to set apart dormant and germinating conidia is the degradation of internal trehalose by trehalases during the breaking of dormancy [175,176]. Subsequently, during isotropic growth, trehalose biosynthesis is activated [69,176]. Trehalases such as TreA and TreB make up a group of enzymes that break down trehalose and together with trehalose biosynthesis transcripts (encoding for TpsA, TpsB, TpsC), TreA-encoding RNA was found to be abundantly present in dormant conidia of *A. nidulans* and *A. niger* [164,170]. After 2 h, during isotropic growth, transcripts of genes involved in the formation of trehalose and trehalase were found to be decreased, but the expression was elevated increasingly throughout the germination [170]. It was found that TreA is either synthesized or activated after germination starts but is less abundant than TreB in *A. fumigatus* [176,177]. Mutated TreB in *A. nidulans* was found to result in a delayed germination phenotype, especially when carbon resources are scarce [176]. 

The transition between dormant and isotropic growth occurs as a response to external triggers such as carbon and nitrogen sources [178]. Dormant conidia can detect molecules in their environment, some of which are able to trigger or forestall germination. To sense the available nitrogen, one of the key resources Aspergilli require to germinate, Aspergilli were found to use G-protein coupled receptors (GPCRs) [178]. *GanB* is an essential gene encoding for the Gα-subunit of a GPCR that induces germination in some Aspergilli, such as *A. nidulans* [179]. Deletion mutants of *ganB* showed a delayed germination phenotype, and although constitutive activation resulted in germination, hyphal growth and asexual sporulation/conidiation were impaired. *GpaB* is the *ganB* homolog in *A. fumigatus* for the Gα-subunit of GPCRs found in *A. nidulans*, but its role in germination was not established [180]. We hypothesize that GpaB might also be important for germination in *A. fumigatus*, although more research is needed to determine whether it can be targeted to halt germination. Notably, other homologous GPCR subunits, such as GpaA and SfaD were found to be compulsory for germination in *A. fumigatus* [181]. 

It was proposed that the immune system clears germinating conidia with exposed β1,3-glucan and α-mannan by Dectin-1 and -2 expressing immune cells, while dormant conidia are bound to airways mucins and macrophages through FleA recognition in a fucose-dependent manner [182]. The lectin FleA (also known as AFL1), was found to play a role in inflammatory (via IL-8 production) [183] and immune response, acting as a PAMP against which the host immune system can act. Its importance in attenuating pathogenicity was proven in a few in vivo studies with FleA-deficient conidia in mice [182,184], highlighting the relevance of the lectin in the proper clearance of conidia. Interestingly, a study [185] showed that bronchial epithelial cells (BEAS-2B) are able to attenuate germination of *A. fumigatus* conidia through recognition of FleA and that this fungistatic activity is effectuated by the phosphoinositide 3-kinase (PI3K) pathway on the host side. The most widely accepted opinion is that FleA recognition by host cells does not endanger internalization; its role in adhesion and phagocytosis, however, is still under debate and its function needs to be elucidated further. In this direction, significant synthetic efforts are currently directed to elaborate strong FleA inhibitors that can probe its involvement in adhesion [186] and filament formation of extracellular *A. fumigatus* [187].

AtfA is an important regulator for the induction of germination in dormant *A. fumigatus* conidia [97]. Earlier, AtfA was found to be a key bZip transcription factor upregulated when conidia experience stress, mediating SakA and MpkC in the MAPK pathway [69]. AtfA expression leads to the upregulation of genes associated with dormant conidia (such as *aspf3*, *aspf8*, *cyp4*, *hsp90*, and *rpL3*), whereas in the absence of AtfA, genes associated with germination were upregulated (such as *calA* and *calB*) [97]. As one of the most upregulated genes shortly after induction of germination, the CalA protein is a proposed germination initiator. What is more, it is one of the genes downregulated in dormant conidia and in the presence of AtfA. Transcription factors regulate many processes within conidia and during host infection, which makes transcription factors such as AtfA interesting targets for novel *Aspergillus*-specific drugs, which were also proposed [67]. Especially seeing that transcription factors tightly regulate gene expression in conidia, which is essential for the different morphotypes of germination and for establishing invasion and colonization of host tissue. 

During the transition from dormant conidia to isotropic growth and polarized growth, the cell wall is remodeled, and transcriptomics studies reflected this with findings that genes associated with biosynthesis of mannose and β1,3-glucan, such as *gel1*, *gel4* and *srbA* are upregulated [173,188,189]. Genes that modulate cellular growth, metabolism and genomic replication are upregulated mostly during polarized growth, in conjunction with cell wall biosynthesis- and remodeling-associated genes, such as *gel2 and gel5* [71,173]. One of the most distinct external hallmarks of polarized growth is the formation of a germ tube, which the host immune system can detect and halt in immunocompetent patients, but not in immunocompromised patients [86]. 

## 5. Proposed Targets for Novel Approach to Aspergillosis Treatments through Targeting Conidial Germination

### 5.1. To Target the Host or the Spores? 

Seeing that many aspergillosis patients are immunocompromised, the question arises whether novel antifungal drugs should target patients, fungal structures such as conidia, or both. The current drugs target specific mechanisms of the pathogens but were proposed to balance the Th1/Th2 immune response, especially for treating ABPA [100]. They proposed that IFNγ might increase the Th1 response and lessen the fungal burden, which is supported by findings that natural killer cells damage germinated conidia by releasing IFNγ in response to *A. fumigatus* [104]. This is also in line with the findings that CF patients exhibit an inflammatory Th2 response, especially when they have ABPA [49]. Mutations in CFTR cause CF, and it was found that CFTR is a key regulator of CD4+ T-cells. Consequently, these mutations in CFTR lead to a bias towards a Th2 response to *A. fumigatus* antigen exposure, as IL-4, IgE and IgG are highly elevated compared to non-immunocompromised people. 

What is more, several other host factors and mutations were found to contribute to a higher fungal burden, as *A. fumigatus* showed more adhesion, internalization, growth, and germination in immunocompromised patients. Firstly, ZNF77 is a transcription factor in humans that regulates extracellular matrix proteins among others, and genetic variant rs35699176 was found to contribute to a higher fungal burden in ABPA patients [190]. Secondly, mutations in the serum amyloid P component (SAP/*APCS* gene) are associated with impaired recruitment of neutrophils upon recognition of conidia [191]. In 2019, seven more host markers were found to facilitate the internalization and processing of conidia by host cells [192]. They proposed the pathways mediated by RAB5C, PIK3C3, and flotillin-2 as potential targets for host-targeted therapeutics.

Although targeting the host rather than the pathogen can be beneficial when the immune system has a high response, patients would still require a responsive immune system. Since CF patients are immunocompromised but are still able to develop an immune response in most cases, they could benefit greatly if the Th1/Th2 balance would shift to Th1. Not all aspergillosis patients are able to produce an immune response, especially immunodeficient patients, and a novel antifungal that targets the Aspergilli rather than the hosts would be more beneficial in these cases. Moreover, it is expected that such an antifungal would probably benefit neutropenic, immunodeficient, and immunocompromised patients, as well as immunocompetent patients (with underlying conditions such as CF and asthma). Antifungals that inhibit germination might be of use as prophylactic agents to prevent fungal outgrowth in patients that subsequently receive immunosuppressive therapy. Except prophylactic treatment should be carefully considered in the face of potential resistance.

### 5.2. Possible Targets for Novel Antifungals That Hinder Germination

Germination is an essential and consistent occurrence in cases of Aspergillosis. As each morphotype exhibits an expression of different genes, and switches within RNA profiles, proteins, metabolism, and cell wall composition were reported [21,71,97,170,173,174,193,194,195]. Proposed potential targets to modulate *Aspergillus* germination can be found in Table 2 and Table 3.

#### 5.2.1. Targeting Dormant Conidia

Dormant conidia of Aspergilli are protected by several cell wall layers, which protect against extracellular stressors and hinder immune recognition of PAMPs as well [86]. As a result of their survival and virulence characteristics, dormant conidia are challenging to target for the host immune system. However, dormant conidia have an extremely distinct transcriptome compared to other germination morphotypes [174]. Upregulated genes in dormant conidia that are possible targets are shown in Table 2; most genes are associated with metabolism, stress resistance, or synthesis of cell wall components. Especially the cell wall and surface proteins in the melanin ‘black-cluster’, along with Aspf3, and Aspf8 are components that are expected to be exposed on the outer surface [173,195]. AtfA is a transcription regulator found to perform a key role in keeping conidia dormant, downregulating genes associated with breaking of dormancy such as *calA* and *calB* [97,208]. Considering the survival mechanism dormant conidia present, affecting cellular components involved in the regulation of internal and external stressors could disturb the conidia enough to reside in a dormant state. Conidia are not completely impenetrable when they are dormant but targeting components that reside within conidia might be challenging as the conidial cell wall is dense and thick. Targeting dormant conidia might still be useful, but rather when conidiation takes place outside hosts, as a means to prevent new conidia from germinating in the environment. Especially because many aspergillosis patients are treated and reside in hospitals for extended periods of time. 

#### 5.2.2. Hindering the Breaking of Dormancy

Conversion of conidia from a dormant state to isotropic growth is facilitated by tight regulation of transcription factors such as Ace2, AmyD, AreA, and NirA (see Table 2) [68,178,200,201]. The transcription factors associated with this morphotype often regulate cell wall remodeling, as is expected, seeing the subsequent morphotype is characterized by conidial expansion. Other potential targets include thaumatin-like proteins CalA and CalB, which are negatively regulated by transcription factor AtfA in dormant conidia [97]. Other than that, abundant transcripts are associated with mitochondria and respiration, fermentation/glycolysis, the TCA cycle, and nitrogen metabolism [174]. Lastly, FleA transcripts are abundant in this morphotype. Since FleA is expressed as a conidial surface protein that mediates adhesion to the host, it can be targeted by anti-adhesive agents designed to prohibit the long-lasting attachment of the microbe on the cell surface, therefore preventing infection [185]. FleA was also found to be upregulated during the transition into isotropic growth [182]. In an alternative approach, FleA expression could be regulated in such a way that the lectin is continuously presented to epithelial host cells, so they can exert their antifungal activity on extracellular conidia through the PI3K pathway. However, additional experiments are required to investigate whether extracellular *A. fumigatus* conidia expressing FleA are inhibited during isotropic or polarized morphotypes. The morphotype is unknown as Richard et al. (2018) incubated the conidia with epithelial cells for 6 h with a reported 15.9%  ±  2.1 germination rate, whereas the conidia incubated with a PI3K inhibitor had a germination rate of 42.7%  ±  10.4 [185]. They did not describe whether these rates corresponded to the formation of a germ tube or whether they checked the conidia for isotropic growth. Nevertheless, the involvement of FleA in cellular recognition and its multifaceted activity makes it a promising target, although (most likely) limited to the *A. fumigatus* or *Aspergillus* conidia. 

#### 5.2.3. Targeting Cell Wall Remodeling That Facilitates Isotropic Growth 

Normally, conidia of *A. fumigatus* expose β-1,3 glucan and other PAMPs during germination, as a consequence of cell wall remodeling that takes place to facilitate conidial expansion and germ tube formation in the isotropic and polarized growth morphotypes, respectively. This remodeling of the cell wall during isotropic growth is facilitated by proteins involved in the metabolism of polysaccharides, such as VadA in *A. nidulans*, ChiA1, and Gel4 together with Gel1 in the Gel family involved in the elongation of β-1,3 glucans in *A. fumigatus* (see Table 2) [173,204]. By targeting the conidial cell wall remodeling, premature germination of conidia could possibly be triggered. This might lead to depletion of energy resources, earlier exposure, and recognition by the host’s immune system, as well as lowering the hypoxial, mechanical, and chemical stress tolerance of the conidia. On the other hand, if essential cell wall synthesis genes such as *gel4* are overexpressed, conidia might be perturbed in their ability to form germ tubes because of a more rigid and near-impenetrable cell wall, hindering or delaying both further isotropic growth and germ tube formation. Disruption of processes where the cell wall is broken down to facilitate the formation and breakthrough of a germ tube could be hindered. ChiA1 is one of these possible targets as it facilitates chitin breakdown [206]. Nevertheless, targeting cell wall breakdown or biosynthesis will promise to be a precarious journey, with reports of Aspergilli mutated in several chitin deacetylases (Cda) with no attenuation of virulence [209].

#### 5.2.4. Targeting Polarized Growth

After isotropic growth, the cell wall was modulated, and as a consequence, the germinating conidia are less resilient to stressors. This is reflected by the transcriptome throughout polarized growth: more transcripts of genes associated with stress resistance such as *ecm33*, *sod3*, and *trr1* are found, as can be seen in Table 2 [27,203,207]. In addition, Ecm33 is not only involved in stress resistance but also in cell wall synthesis and evasion of the immune system. Targeting these stress regulators during the polarized growth morphotype of germination can be conducive to preventing substantial vegetative hyphal growth. Sphingolipids also play a critical role in spore germination and polarized growth and are potential targets [210,211,212]. Cell wall polarity is important for the outgrowth of the germ tube during polarized growth, which is why SrbA is an intriguing target as well [171,213]. SrbA is a membrane-bound transcription factor in the family of sterol regulatory element-binding proteins (SREBPs), that regulates cell wall polarity, and it is involved in the biosynthesis of ergosterol, induced mostly in hypoxic conditions [205,214]. If stabilization does not occur, the conidium is not as capable of rupturing the cell wall with a polar germ tube [205]. Through in vivo experiments in mice lungs, it was found that *srbA* is highly upregulated during infection, highlighting its role in virulence [68]. Gene co-expression analysis was used to identify gene clusters associated with isotropic and polarized growth in *A. fumigatus* conidia. Modules representing highly co-expressed clusters were identified and analyzed [173]. As the conidium grows, the Sienna3 module is upregulated, which contains genes associated with the mitotic spindle and metaphase plate [173]. Even though this is a possible target, it can be anticipated that by targeting structures as conserved as microtubules, host cells are targeted as well as conidia which would be undesirable. Lastly, the bisque4 module is associated with cellular growth, with specific Gel family gene expression of *gel2* and *gel5* in the polarized growth morphotype, with upregulation of *gel1* and *gel4* genes during both isotropic and polarized growth [71,173,215]. Specifically, *gel4* was found as an essential putative glucanosyltransferase in *A. fumigatus* conidia germination [215], whereas *gel1* disruption was not found to impact germination or viability [216].

**Table 3 jof-08-00758-t003:** Other potential targets to disrupt the germination of *Aspergillus* conidia.

Category	Potential Target	Description	Which *Aspergillus*	Source
**Other**	AcuM, AcuK	Key transcription factors associated with gluconeogenesis and acquisition of iron.	*A. fumigatus*	[217]
	CrhB, CrhC	Associated with swelling, germ tube formation and branching. Expressed mostly between t = 1 h and t = 6 h.	A. niger	[218]
	FacB	Transcription factor that is associated with acetate metabolism.	*A. fumigatus*	[219]
	HbxB	Key transcription factor, associated with repressed transcription of genes associated with β-glucan degradation.	A. nidulans	[220]
	MybA	Transcription factor that affects conidial viability.	*A. fumigatus*	[196,221]
	RlmA	Transcription factor that regulates mycotoxin production in conidia, as well as cell wall remodeling and synthesis, in particular, chitin. Associated with the fungal burden in lungs in vivo (mice).	*A. fumigatus*	[68,194]
	TreB	Trehalase, breaks down trehalose during germination.	A. niger	[222,223]
**Hypoxia**	Cox5b, CycA, Afu3g06190, Afu1g1078, Gel4, and Rip1	Most upregulated genes during hypoxia found in vitro with A549 cells.	*A. fumigatus*	[203,205]
		Electron transport chain: complexes III and IV are essential for adaptation under hypoxic growth.	*A. fumigatus*	[224]
	SrbA	Transcription factor in the family of sterol regulatory element-binding proteins (SREBPs). Regulator of cell wall polarity and sterol-associated genes. Involved in iron sensing and adaptation to hypoxia.	*A. fumigatus*	[68,205,213,214]
**Oxidative damage**	CatA, Cat2, Sod3	Superoxide dismutase and catalases, associated with protection from reactive oxygen species and oxidative damage in vitro with A549 cells.	*A. fumigatus*	[203]
	ThiJ/Pfp1 family protein (AFUA_3G01210)	Within the Thil/Pfp1 family. Possibly associated with defense against reactive oxygen species due to similarity to YDR33C in yeast.	*A. fumigatus*	[195]
**Stress response**	AtfA-D	bZip transcription factors, associated with regulation of osmotic and cell wall stress. All four interact with MAPK Saka in conditions that lacked stress.	*A. fumigatus*	[208]
	DprA, DprB, Scf1	Highly upregulated in vitro with A549 cells.	*A. fumigatus*	[203]
	MsbA	Associated functions are within the cell wall integrity pathway, cell wall morphogenesis, and sensor/signaling. A homolog of MSB2 in C. albicans, S. cerevisiae, and A. nidulans, functional as an external sensor and important for virulence.	*A. fumigatus*	[225]
**Downregulated after breaking dormancy**	Cat2, MirD, Sdh2, SidA, SidC, SidD, SidF	Associated with iron acquisition. Downregulated in vitro with A549 cells. Only cat2 sidA, sidD, and mirD were found to be downregulated in vivo [68].	*A. fumigatus*	[203]
	GpgA	GPCR-γ subunit associated with gliotoxin production. A loss of function mutant showed severely delayed and impaired germination, with reduced structures in the maximum 65% germinated conidia.	*A. fumigatus*	[42,181,203]
	SltA	Downregulated as a response to nutrient deficiencies during growth in vivo (mice, IPA model).	*A. fumigatus*	[68]

#### 5.2.5. Other Anti-Germination Targets 

Affecting the components involved in stress resistance of conidia in dormant stadia and during early germination would impede vegetative growth of hyphae. This would prevent penetration and invasion of host tissue, resulting in a decreased fungal burden for patients. Hypoxia was identified as a stressor for *A. fumigatus* and several genes upregulated during germination are associated with hypoxia stress tolerance, as displayed in Table 3. Upon contact with host cells, *A. fumigatus* downregulates genes associated with virulence such as *sidA*, *sidD*, and *gpgA* in conidia [68,203]. This could contribute to the evasion of the host’s immune system, by adapting to the host’s environment [84]. 

## 6. Discussion and Conclusions

Aspergillosis has a tremendous impact on humanity, in particular the neutropenic population, as a result of congenital defects [226], infections with pathogens [227,228,229], or following treatment [230,231]. The latter is most often the case due to corticosteroid therapies [73,232] and chemotherapeutic agents [233,234,235]. Not only are Aspergilli the cause of disease in over 14 million people worldwide, but they cause high mortality rates depending on the specific disease [3]. Invasive infections have reported mortality rates of 30 to 95% [3], whereas chronic aspergillosis has minimal mortality of 15% in the first 6 months following diagnosis [2]. Additionally, Aspergilli possibly contribute to a higher susceptibility to other pathogens as a consequence of the burden they pose on patients [36]. 

Conidia are the most important causative agents of these infections since they are dispersed easily and once inhaled reach deep parts of the lungs. *A. fumigatus* was found to adapt to its surroundings by delaying germination and by modulating gene expression, upon recognition of neutrophils or epithelial cells [84,98,200]. As a direct consequence of delayed germination, *A. fumigatus* conidia are affected less by macrophages and neutrophils compared to *A. niger*, along with reduced immune response and more efficient internalization by lung epithelial cells [84,203]. The host immune system is an important factor to consider, especially combined with the ability of Aspergilli to evade the immune system. A course of action would be to negate the interplay of aspergillosis and the immune system, by modulating or correcting the immune system of the host. However, this is not a possibility for immunodeficient patients, as the state of the immune system is often already modulated through treatment. Patients undergoing corticosteroid treatments or chemotherapy are often chronically or temporarily neutropenic. Therefore, modulating their deficient or malfunctioning immune system to combat Aspergillosis would not benefit the patients considerably. Similarly, other susceptible groups have no prospective or currently effective treatments. Thus, solely modulating the host is not desirable as it does not prevent aspergillosis for many susceptible patients. Moreover, immunocompetent patients with COPD and PTB, among others who suffer from aspergillosis would not benefit from this specific type of treatment. IFNγ was tested as adjunctive immunotherapy for invasive fungal infections [236], but one should also consider the adverse effects of IFNγ that were previously reported [237]. Another consideration is the polymicrobial environment the lungs of many *Aspergillus*-affected patients have, as it was found to have an impact on the efficacy of antimicrobial agents [238]. Therefore, another approach that specifically targets Aspergilli would be pertinent for all aspergillosis patients. 

Current treatments to combat aspergillosis consist of mainly antifungal azoles such as voriconazole, but there has been a rising resistance against azoles since 1998/1999 [19,62,122]. Consequently, the mortality of aspergillosis was found to be about 88% for azole-resistant *A. fumigatus* infections [160]. Most concerning is that azole-resistant *A. fumigatus* can be found quite often, with a resistance prevalence as high as 7.1% in a study of CF patients [63] and 11.3% in IPA patients [239]. In the search for novel antifungal drugs, it is essential to consider factors that facilitated azole resistance, as a means to prevent the reoccurrence of drug resistance. Altogether, this prompted the main question and aim of this review; to combat aspergillosis, what are potential targets to preclude *Aspergillus* spp. germination?

The approach to determine novel targets to hinder germination was to review and inspect transcriptomic and proteomic studies on conidial and hyphal gene expression. We feel that it is relevant to remark upon the difference between in vivo and in vitro studies; a study found that the transcriptional profiles of *A. fumigatus* in vivo were considerably different, with higher induction of gene expression, up to 1200 genes more when compared to in vitro assays without human or murine cells [217]. Several studies have identified in vivo transcriptomic profiles of *A. fumigatus* conidia during infection of host lung epithelial tissue [42,68]. However, most studies with human cells were conducted in vitro with primary human airway epithelial cells [39], lung epithelial cells A549 [84,203], 16HBE bronchial epithelial cells [39,190], blood [200], or neutrophils [98]. One study used a specific A549 type II pneumocyte cell line, as well as human polymorphonuclear leukocytes (PMNs) and monocyte-derived dendritic cells (moDCs), from healthy donors [83]. Another important aspect to consider is that even though all these experiments were conducted with *A. fumigatus* and most use the Af293 isolate, others use specific clinical isolates. The repercussion is that the findings of these studies might not be fully comparable. All in all, there are some discrepancies among these studies, but they are important to consider gene expression during germination. 

Firstly, some of the potential targets are expressed exclusively in a certain morphotype of conidial germination, whereas other targets are not distinct for one morphotype, but are still specific for germination. Dormant conidia and the breaking of dormancy are compelling morphotypes to target during conidial germination, but due to the tight regulation involved in the breaking of dormancy, might be too confining for a novel antifungal. Especially, seeing that for instance transcripts of transcription factor Ace2 were found to be upregulated after incubating for half an hour, but downregulated after 2.5 h total incubation time [200]. FleA might be one of the most promising targets for blocking the breaking of dormancy, as it does not depend on normal host immune responses based on neutrophils, but rather on epithelial cells [185]. Thus, treatments targeting FleA in neutropenic patients with *A. fumigatus* infections might facilitate the clearance of an early infection by themselves. At the same time, immunocompetent patients with aspergillosis would benefit from a targeted halt of conidial germination. Constitutive FleA on conidia might be targeted through the promotor but blocking FleA degradation potentially has a similar effect. We used NCBI BLASTN to examine whether FleA homologs of *A. fumigatus* Af293 are present in other pathogenic Aspergilli (Gene ID: 3511258). Consequently, all five Aspergilli aligned with the query of pathogens: *A. pseudoviridinutans*, *A. fischeri* NRRL 181, *A. novofumigatus* IBT 16806, and *A. lentulus*. Even though it seems to be the case that not all pathogenic Aspergilli can be targeted through FleA, it seems to be a conceivable optional target for at least the primary strain causative of aspergillosis (*A. fumigatus*). Promisingly, a BLASTP (Accession XP_753183.1) resulted in a high query coverage of the amino acid sequence of FleA with more Aspergilli than the search with BLASTN, including pathogenic species such as *A. flavus*. 

Next to Aspergilli, *Scedosporium* spp. is the second-most prevalent fungus in the lungs of CF patients [240] and a homolog of FleA has recently been found and characterized in *Scedosporium apiospermum* [202]. Findings of homologs of FleA in other fungi that cause mycosis are promising for the impact potential FleA-targeting antifungals could have on the battle against fungal infections.

Both isotropic and polarized growth are associated with a higher susceptibility to stressors and more recognition of PAMPs by the host immune system. Hence, it is compelling to utilize the native vulnerabilities by targeting these genes, transcription factors, and proteins. Nonetheless, a conceivable downfall of this approach is that there are many presumed redundant genes, especially within stress resistance. Therefore, mutation-driven resistance is more probable to occur, as has already been demonstrated for SrbA and Ace2 [139,159,213]. By targeting several factors within stress resistance-associated genes, the redundancy might be bypassed, resulting in a lower chance of mutation-driven resistance.

We propose to target several essential components within conidial germination in antifungal treatment, as this would limit the likelihood of a single escape mutant acquiring resistance and colonizing patient tissue. Combination therapy was found to be more efficacious for several applications of drugs targeting mutating targets, such as in cancers [241,242,243] or Gram-negative bacteria [244]. Considering that azole use by agricultural and commercial sectors has contributed to azole resistance it is essential to prevent the application of structurally related fungicides [153]. The emergence of medically induced in-host-acquired azole resistance could be restricted through recently improved diagnostics, along with the detection of mutations indicating resistance [245,246]. 

The proposed targets can be evaluated by determining (protein) structure, localization in (germinating) conidia with confocal microscopy, as well as studying interactions. Tools such as STRING could contribute since they display networks of known and predicted protein–protein interactions. By using STRING, the *A. fumigatus* Gel4 protein was found to interact with Ecm33, Crf2, ExgA, and some other unnamed proteins (complete interaction network shown in Figure 3) [247]. These insights can contribute to a well-rounded estimation of the consequences of targeting certain genes. Several targets were experimentally tested, although an approach with loss-of-function, overexpression, or SNP mutants, with a specific focus on germination of conidia, would be called for, specifically to analyze whether affecting the proposed targets halts or delays certain germination morphotypes as is predicted. Others were only tested in culture, on different media, and thus different stressors and/or energy and nutrient limitations. Others have not been tested in the presence of epithelial cells in vitro, or in vivo, while both would provide more insight relevant to patient treatments.

In this review, we propose to target specific morphotypes of conidial germination for the development of novel antifungal treatments for aspergilloses. This is mainly aimed at neutropenic patients, where azole resistance emerges frequently, and current treatments are severely insufficient. We think that immunocompetent patients could perhaps benefit more from antifungals that target conidiation [66]. The antifungals could have a possible prophylactic application for patients at the start of, e.g., chemotherapy, to completely obstruct *Aspergillus* growth. As antifungals would specifically target Aspergilli, one could expect lower patient toxicity than current antifungals pose. Personalized treatments, dependent on immune status, could offer a means to prevent resistance as well as both specific and broad activity. The primary drug could be a novel antifungal targeting germination, which would provide a broad activity against Aspergilli. This could be made to target several genes, proteins and/or transcription factors within one or several germination morphotypes. A secondary drug could enhance or complement the immune response in immunocompetent patients, wherein the immune system could be balanced in, e.g., ABPA/CF patients, and possibly lead to enhanced clearance by bronchial epithelial cells in immunodeficient patients. To sum up, combination therapy of novel antifungals targeting germination could mainly benefit neutropenic patients affected by aspergillosis.

## Figures and Tables

**Figure 1 jof-08-00758-f001:**
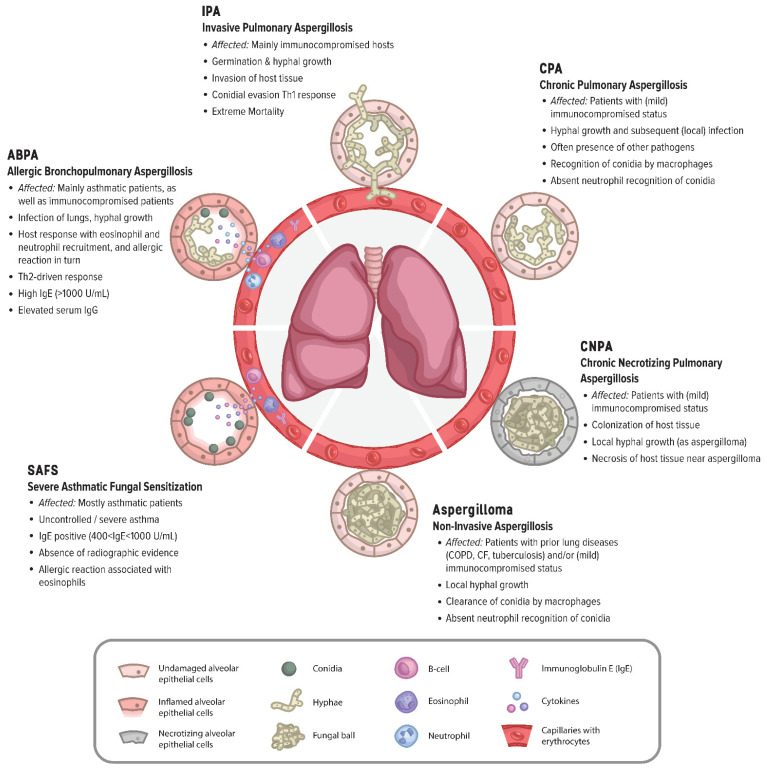
Schematic overview of common lung diseases caused by Aspergilli. Representative alveoli, capillaries with red blood cells and immune cells involved in the direct pathology of aspergilloses such as allergic bronchopulmonary aspergillosis (ABPA) and severe asthma with fungal sensitization (SAFS).

**Figure 2 jof-08-00758-f002:**
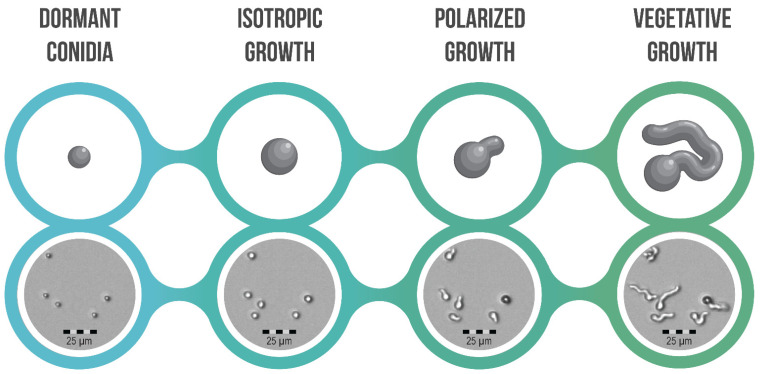
Morphotypes of conidial germination in Aspergilli. Graphical representation (**top**) and Fluidscope^TM^ images of *A. fumigatus* conidia taken through time with oCelloScope (**bottom**) (Biosense Solutions, www.biosensesolutions.dk, accessed on 21 July 2022 [172]).

**Figure 3 jof-08-00758-f003:**
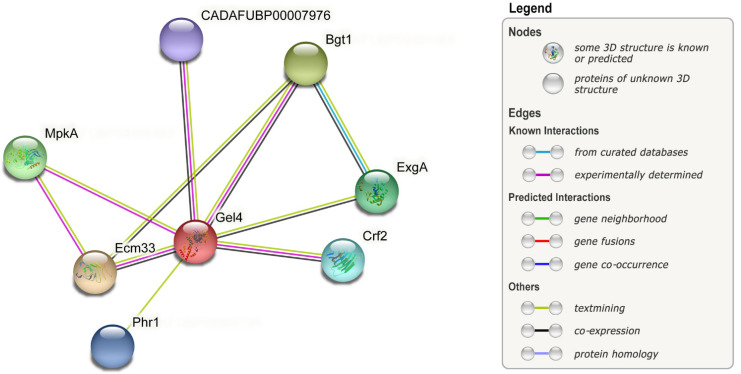
Protein interaction network of query protein Gel4 (red node) in *Aspergillus fumigatus* (*A. fumigatus*) (NCBI taxonomy Id: 746128) with reduced legend. Each node represents a protein. Filled nodes have a known or predicted 3D structure, whereas empty nodes do not. Edges represent protein–protein interactions and colors indicate known, predicted, or other interactions. The protein CADAFUBP0000 7976 is unnamed and has the protein accession code: XP_747364.1 Produced with the STRING tool (V11.5) [247] by K. Verburg.

**Table 2 jof-08-00758-t002:** Potential targets to disrupt the specific morphotypes of germination of *Aspergillus* conidia.

Germination Morphotype to Disrupt	Potential Target	Description	Which *Aspergillus*	Source
**Dormant** **conidia**	Arp1, Arp2, Ayg1	Associated with melanin biosynthesis (black-cluster Baltussen et al., 2018). Expression was found to be exclusively high in dormant conidia.	*A. fumigatus*	[173,195]
	Aspf3, Aspf8	Cell surface-associated proteins can be recognized as allergen by host’s immune system.	*A. fumigatus*	[97]
	AtfA	Key transcription factor present in dormant conidia that negatively regulates calA and calB, which are involved in breaking of dormancy.	*A. fumigatus*	[97,196]
	CatA, ConJ, Fhk1	Genes that are upregulated in an AtfA-dependent manner. CatA is a spore-specific catalase. ConJ has an unknown function in *A. fumigatus*. Fhk1 is a transcription factor that regulates the CLB2 cluster of genes in the G2/M phase of the cell cycle, associated with cell growth, mitosis, and cytokinesis.	*A. fumigatus*	[97]
	CatA, Cat2, Cat3	Catalases that protect dormant conidia against oxidative stress.	*A. fumigatus*	[97,195]
	CcpA	Associated with stress resistance in vitro with cells.	*A. fumigatus*	[83]
	Cyp4	Peptidyl-prolyl cis-trans isomerase.	*A. fumigatus*	[83]
	DprA, DprB, DprC	Dehydrin-like proteins involved in stress–response of dormant conidia, upregulated in an AtfA-dependent manner.	*A. fumigatus*	[83]
	Hsp90	Heat-shock protein, associated with temperature stress.	*A. fumigatus*	[83,166,197,198,199]
	RpL3	Ribosomal protein L3.	*A. fumigatus*	[83]
		Involved in alcohol fermentation (pyruvate decarboxylase and alcohol dehydrogenase).	*A. fumigatus*	[174,195]
**Breaking of Dormancy**	Ace2	Transcription factor for Swi5, regulates germination, pigment production, and virulence. Tightly regulated, upregulated at t = 0.5 h, downregulated at t = 2.5 h.	*A. fumigatus*	[200]

	AmyD	Key regulator associated with α-glucan synthesis and cell wall remodeling.	A. nidulans	[201]
	AreA, NirA	Transcription activators that respond to nitrogen. Found to be germination triggers and for nitrogen uptake	*A. fumigatus*	[68,178]
	CalA, CalB	Thaumatin-like protein, associated with triggering breaking of dormancy. Negatively regulated by AtfA.	*A. fumigatus*	[97]
	CreA (An02g03830), (An02g03540)	Fermentation/ Glycolysis: creA is a catabolite repressor.		[174]
	FleA	Recognizes and binds receptors, plays a role in attachment/adhesion to epithelial cells, as well as recognition by host’s immune system.	*A. fumigatus* and *S. apiospermum* (as SapL1)	[182,183,184,185,186,187,202]
	PmaA, (An11g04370), (An01g10190), (An04g02550), (An08g08720)	Mitochondria/Respiration.	*A. fumigatus*	[174]
	-Translation initiation factor CpcC -Transcription factor CpcA -Neutral amino acid transporters (An16g05880, An04g09420, An17g00860) -Transporter proteins (An11g00450), (An03g05590)	Nitrogen metabolism.	*A. fumigatus*	[174]
		TCA cycle.	*A. fumigatus*	[174]
**Isotropic growth**	Gel1, Gel4	Gel family, important for cell wall remodeling. Linking and elongating of β-1,3-glucans.	*A. fumigatus*	[173,203]
	VadA	Transcription factor involved in regulation of genes associated with polysaccharide metabolism, cell wall, and stress.	*A. nidulans*	[204]
**Polarized growth**	Bisque4 module	Associated with cellular growth, includes genes such as sun1 (involved in modification of β-1,3-glucan), sidA (essential for the primary step of siderophore biosynthesis), GEL family genes (gel2, gel3, gel5), and chitin synthase genes.	*A. fumigatus*	[170,173]
	ChiA1	Class III chitinase, associated with conidial stress, upregulated in hypoxic conditions.	*A. fumigatus*	[205,206]
	Ecm33	GPI-anchored protein associated with cell wall biosynthesis, stress resistance, and evasion of host’s immune system.	*A. fumigatus*	[27,207]
	Sienna3 module	Associated with regulation of the cell cycle and DNA processing -mitotic metaphase plate congression -assembly of the midzone of the mitotic spindle -nucleation of microtubules by the spindle pole body.	*A. fumigatus*	[173]
	SrbA	Transcription factor in the family of sterol regulatory element-binding proteins (SREBPs). Regulator of cell wall polarity and essential for outgrowth of germ tubes.	*A. fumigatus*	[171,205]
	Sod3	Sod3 is an allergenic putative manganese superoxide dismutase, associated with reactive oxygen defense.	*A. fumigatus*	[203]
	Trr1	Putative thioredoxin reductase.	*A. fumigatus*	[203]

## Data Availability

Not applicable.

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
