# Peer review of "Novel Treatment Approach for Aspergilloses by Targeting Germination"

_jof, 2022, doi:10.3390/jof8080758_

Round 1
Reviewer 1 Report
The proposed review by Verburg and colleagues focuses on Aspergillus spore germination and host-pathogen interaction. This is an interesting topic because germination is an essential and early step in Aspergillus infection in the lungs. In addition to the introduction, the authors have divided their discussion into 4 main themes. Host susceptibility, current treatments and azole resistance; conidial germination and its different morphotypes; and finally, the proposed target for a new approach to the treatment of aspergillosis by targeting conidial germination.
Main comments:
Many antifungal drugs have been identified. Only few of them are used in clinics. It would be interesting to add other known antifungal targets/drugs that are in clinical trials in Table I. GPI anchor biosynthesis or sphingolipid synthesis as examples (see https://doi.org/10.3390/jof6040213).
In “Targeting cell wall remodeling that facilitates isotopic growth and targeting polarized growth”, this part is mostly restricted to Gel family and would deserve to be more complete. Of the cell wall polysaccharides in A. fumigatus, 3 have been described as essential for fungal filamentation and polarized growth: β-1,3-glucan, chitin, and galactomannan. GM is required for conidiation and regulation of polarization (see papers from Latge's group), indicating that GM synthesis (GDP-Man transporter, mannosyltransferases) and its translocation (DFGp) into the cell wall represent antifungal targets as chitin and β-1,3-glucan remodeling.
Sphingolipids also play a critical role in spore germination and polarized growth (Levery et al., FEBS, 2002, 525:59-37; Li et al. 2007, Genetics 176: 243-253; Li et al. Mol Cell Biol 2006, 17:1218-1227). Inhibition of sphingolipid synthesis (glucosylceramide or inositolphosphoceramide) blocks the early stage of germination. Furthermore, in chronic pulmonary inflammatory diseases such as asthma or cystic fibrosis, Aspergillus infection increases de novo sphingolipid synthesis, which leads to the accumulation of ceramides and the mediation of the inflammatory response in the lung tissue. Interestingly, inhibition of serine palmitoyltransferase by myriocin has a dual anti-inflammatory and antifungal effect (Caretti et al. BBA 1860 (2016), pp. 1089-1097). Lipid metabolisms may be worth discussing in the review.
The latter topic (targeting conidial germination) remains too speculative. Diagnosis of Aspergillus infection using biomarkers (galactomannan, β-glucan, PCR) or clinical symptoms comes later in the infection, when vegetative growth is well established. Germination blockade appears to be an inappropriate or only prophylactic antifungal target. In Table 2, a number of putative targets have been cited, but not all discussed in the text. For example, how might melanin biosynthesis be an antifungal target? It would be appreciated if the authors could better explain their purpose and perspective in this section. Spore germination is the first step in Aspergillus infection that leads to the initial interaction between host and pathogen, early immune response and adaptation of the fungus to the host environment. It would be appreciated that authors address this important topic in more detail.
Minor comments:
Line 148: GAG is absent from the dormant conidia. This cell wall polymer is produced during germination and vegetative growth.
Line 373 : Transcripts of trehalose? sentence to rewrite.
Author Response
please find our response int he attached file

Reviewer 2 Report
Dear authors,
I read your manuscript concerning the Novel treatment approach for aspergilloses by targeting germination with interest. The paper is well-written and structured, figures are clear and easy to understand. Moreover, tables are necessary for the readers. I report some notes to improve the paper’s quality:
1) In all the text, you don’t mention cryptic Aspergillus species, representing a current and difficult to treat the araising problem. Read and cite:
- Tsang CC, Tang JYM, Ye H, et al. Rare/cryptic Aspergillus species infections and importance of antifungal susceptibility testing. Mycoses. 2020;63(12):1283-1298. doi:10.1111/myc.13158
- Rozaliyani A, Abdullah A, Setianingrum F, et al. Unravelling the Molecular Identification and Antifungal Susceptibility Profiles of Aspergillus spp. Isolated from Chronic Pulmonary Aspergillosis Patients in Jakarta, Indonesia: The Emergence of Cryptic Species. J Fungi (Basel). 2022;8(4):411. Published 2022 Apr 16. doi:10.3390/jof8040411
2) Line 125-130. It looks as if only immunocompromising plays in the peculiar infection. You should report that also host genetics (dectin1/2; STAT3 mutation) contribute to Aspergilli infection.
3) Line 186, italic style, please
4) Line 225-228. Other than P. aeruginosa, Achromobacter spp. is emerging as bacteria involved in FC patients. Did you know any relation between A. xylosoxidans and fungal infection?
5) Line 232-234. You mention the TLR5-mediated switch. Did you think anti-IgE monoclonal therapies, such as Dupilumab, could be useful in those clinical manifestations or in atopic diathesis patients?
6) Did you think iron chelators should be mentioned in your paper? Explain.
7) Line 437-439. INF-g was largely used in other infections, such as HCV. The most limiting factor was/are adverse events in patients. Report these limitations with adequate references.
8) Line 456-458. Immunotherapy is not cited in the text, but it is a milestone in oncological daily routine, especially in melanoma and lung cancer. You must include some lines about the rational use of your revised target and what should be the gold standard in this key population (immunotreated)
9) HSP90 is one of the current targets in Aspergillus spp. Read and cite the following papers to underling its importance and retinoids, as novel agents in fungal infection:
- Cosio T, Gaziano R, Zuccari G, et al. Retinoids in Fungal Infections: From Bench to Bedside. Pharmaceuticals (Basel). 2021;14(10):962. Published 2021 Sep 24. doi:10.3390/ph14100962
- Campione E, Cosio T, Lanna C, et al. Predictive role of vitamin A serum concentration in psoriatic patients treated with IL-17 inhibitors to prevent skin and systemic fungal infections. J Pharmacol Sci. 2020;144(1):52-56. doi:10.1016/j.jphs.2020.06.003
- Campione E, Gaziano R, Doldo E, et al. Antifungal Effect of All-trans Retinoic Acid against Aspergillus fumigatus In Vitro and in a Pulmonary Aspergillosis In Vivo Model. Antimicrob Agents Chemother. 2021;65(3):e01874-20. Published 2021 Feb 17. doi:10.1128/AAC.01874-20
Author Response
please find our response in the file

Reviewer 3 Report
The manuscript Jof-1787910 reviews the literature around germination in Aspergilli to highlight novel treatment approaches to aspergilloses. The review is clear and well-written, and I would like to commend the authors on their excellent figures.
There is an appropriate amount of literature review gone into this manuscript. However, several key aspects of conidial germination have not been discussed. In addition, I would like to highlight several small issues before this manuscript is ready for publication.
Major comments:
- The manuscript does not discuss melanin at all. Cell wall and conidial components are discussed in much detail, so it strikes me that this is being left out. It is such a key component of fungal recognition through MelLec.
- Conidial recognition by the immune system is mentioned throughout the manuscript and how dormant conidia are not recognised. However, there is a vast amount of data on epithelial cell uptake of spores and killing. Epithelial cells are part of the immune system and should be included.
- Similarly, CF has been mentioned throughout the manuscript. The effect of excess mucus has not been discussed and how this impairs fungal clearance.
Minor comments:
- L11 "Conidia can evade the immune system" Evasion is an active process. Conidia are not detected by the majority of leukocytes.
- L21: "germination of conidia is one of the few common denominators" This is not true, ABPA patients do not require germed conidia to have disease. Hyphal fragments (dead or alive) or cell wall debris can already cause immune activation.
- L30 "Most invasive fungal pathogens" most invasive definitely not, mucor invades the tissue much deeper. If "most common" is meant that is true.
- L39-40 This depends on what country, this data can't be generalised.
- L63: "higher pathogenicity" Higher prevalence and higher infection rates maybe. I don't agree with higher pathogenicity
- L81-82 "by host cells" what cells? Depends on what cell
- L85: "the conidia can colonize the host tissue" Not conidia, the fungus
- L88-89 "co-infection and IPA" co-infection with IPA?
- L104 Other transcription factors have been described; NctA, NctB, HapB, StuA and others. I would recommend some more literature to be included.
- L115-117: Include influenza and COVID-19
- L135: A. niger as example of plant pathogen is not the best. A. flavus is much more common as plant pathogen.
-L138,139: Is not supported by these references. At least the temperature and humidity. Find primary data and references or use meteorological/ecological findings to show that it is present in air/soil at higher temperatures and low humidity.
- L141-142 "Virulence is associated with ability to disperse" I disagree, not virulence itself but how often we are challenged with this fungus.
- L149-152 Others rodlets are present and should also be discussed briefly.
- L186 A. fumigatus in italic
- L197-198 Aspergillus doesn't target immune deficient patients. It is just not cleared and stopped, target sounds like the fungus actively seeks them out.
- L209 "healthy host monocytes A. fumigatus recognition" not sure what is meant here?
- L225 Co-infection of Aspergilli AND P. aeruginosa
- L237 Ketoconazole was first used in the 80s before replaced by itraconazole. However, it is out of use now.
- L239-242 Here is a chance to discuss L-AMB and other formulations that are available, and mostly in use now.
- L252-253 6-12.8% depends on the country, please mention where from and that other countries see other percentages.
- L264 I can mention many more, this is a bit of an odd list. Is this clinically found? HapE, cdr1b, nctA has been found in patients
- L265-267 Introduce TR46 here
- Table 1: I'm missing anidulafungin here
- Cyp51b mutations should also be discussed here
- L295-297 Rhodes et al 2022 has shown through population genomics this is the case.
- L334-337 RNA-seq of this is also available in Danion et al 2021. I would recommend reading the implications of this work on nutrients and breaking of dormancy
- Section 4 is missing heterogeneity within spores. Work mentioned above and work from Bleichrodt et al 2020, Kang et al 2021 and Wang et al 2021 should be discussed.
- L379 There is work that shows that water alone can activate spores. Worth a look.
- L408-421 Why this focus particular on AtfA, I'm missing the relevance as it is a transcription factor involved but not the only one, a full paragraph seems overdoing it.
- Table 2. A. nidulans italic and ChiA1 is missing a species
- L480 Upregulated genes in dormant conidia, I believe they look at transcript levels and don't do differental expression.
- L497: There are many more transcription factors, why are these mentioned and not others?
- L554 and L558: Sienna3 and bisque4 are randomly assigned names for modules. Describe what is in these modules because mentioning these names alone is meaningless.
- Can CADAFUB in Figure 3 be converted to standard in use protein names. In addition, protein names should start with a capital letter.
Author Response
please find our response in the attached file

Round 2
Reviewer 2 Report
Dear Authors,
all the corrections have been made.